# Multi-Faceted Analysis of Airborne Noise Impact in the Port of Split (I)

**Luka Vukić \***[ID]**, Ivan Peronja and Roko Glavinović**

Department for Maritime Management Technologies, Faculty of Maritime Studies, University of Split, Ruđera Boškovića 37, 21000 Split, Croatia
* Correspondence: luka.vukic@pfst.hr; Tel.: +385-98-549-849

**Abstract:** This multi-faceted study deals with the analysis of the impact of noise emissions from the cargo terminals in the port of Split, especially in view of the proximity to inhabited areas and the growing number of registered issues and concerns due to its particular location. Three objectives are pursued: the identification of noise sources in the port area, an overview of strategic noise maps and simulations of noise propagation from ships at berth, and the calculation of external costs of noise pollution. In the first, preliminary part of the research project, by conducting a monitoring campaign and analyzing the data on strategic noise maps of the studied area, road and rail traffic were estimated as the main noise sources causing excessive noise emissions for all assessment periods: day ($L_{day}$), evening ($L_{evening}$), night ($L_{night}$), and day-evening-night ($L_{den}$) period. Industrial resources, including ports, were identified as having marginal noise emission levels. The calculation of the total external noise costs results in a damage value of €190,166/year, considering the number of affected inhabitants and the assumed noise levels. As an added value of the study, the simulation results of two scenarios have determined the noise propagation of a ship at berth and highlighted the zone of excessive noise under certain conditions. The results of this study should encourage the relevant institutions to strengthen noise management plans and introduce effective and continuous monitoring of noise emissions in critical areas.

**Keywords:** airborne noise; port sources; mapping and simulation; external costs

## 1. Introduction

Ports are hubs essential for carrying out logistic and industrial activities within global supply chains [1]. The role and importance of seaports have evolved from the traditional traffic, commercial, and industrial functions to become an inherent link in distribution networks. They have enabled the development of a novel logistic function as a contemporary approach to global challenges in the transportation system, by providing added value to their primary activities and services. Considering the rising port influence and a long-term prediction of stable growth of the shipping industry [2], environmental issues are becoming more significant and progressively decisive in the planning of future commercial activities, and thus, freight distribution. Additionally, as most European port sites are situated in city centers, the pressure of transport activities on local communities and sustainability patterns increases, generating hazardous influence on socio-ecological factors. Environmental noise generated by transport activities is the most prevalent source of pollution, right behind fine particulate matter pollution, when considering environmental impacts on health in western Europe [3–5]. Until recently, most research considering environmental noise health effects has been focused on road, rail, and aircraft noise—due to their similar characteristics, as indicated by Paschalidou et al. (2019) [6]—excluding impacts from maritime and port transport activities. However, the issue of port noise has gained special attention in the scientific community due to the rising number of complaints from people living in urban areas adjacent to ports [7]. The rise of port traffic on a global scale will certainly

coincide with an increase in environmental noise emitted, especially in cases where the implementation of adequate strategy and management of noise-mitigation are missing [8]. Unlike underwater maritime noise impact, examined extensively in the literature [9–16] and suitably regulated [17], airborne port noise has been studied significantly less in the scientific community, resulting in a lack of international regulations and the absence of international standards for noise management [18]. The current regulations classify port noise as industrial noise [19], which neglects the harmful noise exposure of the urban population near ports [3]. This is individually the most significant cause of port noise neglect, as the other transport modalities are included in the European Noise Directive requiring strategic noise-map creation for road, rail, airports, and urban centers [20]. Thus, the limitations which arise from a deficient regulatory framework do not follow the increasing environmental concern, which generates a chain reaction through the other relevant documentation, procedures, and finally, perceptions of the main stakeholders. The current regulatory framework implications, visible in the documents from van Essen et al. (2019) [21], provide an overview of the transport sector's external costs. They indicate that the non-existence and negligence of noise costs in maritime transport is due to the commercial voyage nature, which occurs in sparsely inhabited areas. However, this fact can be applied only when international shipping is considered, and firmly excludes the noise emitted in short-sea shipping, and especially in ports. Considering the prolonged and frequent exposure to noise, the ships in ports generate low-frequency noise propagating in open space at significant distances. The airborne noise generated during the ship's cruising phase is primarily emitted in rural areas, having a lower impact [22]. As for the increasingly harmful noise impact in ports, usually located in populated areas, there is a growing interest in local communities which are directly affected by airborne port noise [9]. Prolonged exposure to environmental noise increases the risk of physiological and psychological damage, such as cardiovascular and metabolic effects, cognitive impairment in children, and severe effects of annoyance and sleep disturbance [23]. There is a comprehensive list of noise health effects in the study by Petri et al. (2022) [24]. Sound pressure levels above 75 dB(A) can cause significant hearing loss, especially if a longer duration of exposure is considered [25]. As for the nature and complexity of the port—which, besides ships at berth as the principal noise source, also comprises numerous different industrial activities and transport operations—it is necessary to define and analyze all the noise sources in the port area. These activities are mainly related to the provision of transport services connected with available cargo-handling equipment and utilization of intermodal transport capacity. It is important to emphasize that these noise sources are later used as inputs and included in the noise-exposure assessment for developing noise and acoustic maps of the referent area. Noise mapping is an effective tool that provides the basic information required to identify sources of noise generated in the port area, while also indicating the individual sources with the highest impact [26]. It is crucial to apply noise-abatement strategies in order to identify the dominant sources of environmental noise, which represents a challenge in a complex port environment [27]. Noise mapping is used for the future planning of port development when considering direct and indirect impacts according to noise standards. It also enables noise management, which intends to minimize the negative implications to the community and environment. The output mode of the acoustic maps defines the areas of noise emission and indicates the number of people exposed to certain noise classes. Significant noise health damage from ships in ports is already reported in some locations. This includes the issues and complaints reported from locals in Dublin and Athens [6,28] and those registered in La Spezia and Nice [29]. The combination of the number of people exposed with the noise costs per person enables the calculation of the total environmental noise costs, which are defined as one of the main externalities in transport [21]. Society compensates for external transport costs rather than the transport providers, and noise pollution costs have significantly increased over the last period, reaching a 7% share of the total external transportation costs [30]. Due to the lack of official directives and guidelines at a regional level and policies at a local level considering noise pollution, along with a

small sample of recorded information and standardized data [31], several civil projects have been implemented. The purpose of this is to support the authorities who are aiming to determine the nature of port activities as potential noise sources with their acoustic features, and provide policy recommendations based on the data generated.

This research aims to classify all the noise sources in the port of Split according to the specific port environment, focusing on the freight terminals where conventional cargo-handling activities are performed. Firstly, the noise sources are categorized as either industrial or those related to transport operations. Afterward, they are divided into several cross-sectors according to their specific characteristics and roles in the port area. This data is used in noise-mapping the port area, presenting valuable resources for decision-makers to direct activities and create noise-management policies and mitigation measures. The second activity relates to the acoustic map analysis and theoretical simulation of noise propagation on the cargo terminal's open surface. This phase enables recognition of the critical points and provides conclusions for the hypothetical situation created in the specific environment of the research problem. The simulation results, in predefined conditions, provide theoretical knowledge which enables comparison of the outcomes in actual measurement units; these will be generated in the further phases of the general project. The final step includes the calculation of external noise costs for the subject area, based on the most relevant literature. As the fundamental objective of noise measurement and validation in the port is to determine the social and environmental effects, an overview of the total damage costs compensated by the local community is provided. The monetization of noise pollution damage shows the magnitude of the expenses generated due to the location of the unwanted acoustic sources.

Based on the set objectives and planned activities, the authors defined three research questions that should allow for drawing conclusions from the presented work:

- According to the conducted monitoring campaign, which noise sources in the port area generate excessive noise levels?
- What is the extent of noise propagation in the port and its surroundings when simulations are applied (to determine the exposure of the local population)?
- What is the magnitude of noise environmental damage from port activities expressed monetarily?

## 2. Literature Review

As already stated, the issues of port noise have not gained significant attention in the academic community until recently. Currently, scholars are focused on theoretical and empirical research considering the extensive scope of noise effects in the maritime industry. This includes topics regarding ship noise from various aspects, like noise on berth, in movement, seafarer's health impact, different locations, and specific conditions. However, most studies still deal with underwater noise as the primary research problem in the maritime industry. These noise effects were excluded from the analysis in this research. The authors reviewed the 12 most relevant scientific journal papers, 6 conference papers, and 2 chapters in a book on airborne port noise, categorizing them as regarding ship noise, noise generated in the port area, and noise in the port system. In addition, the literature review considered seven reports with significant contributions concerning their scope. Most papers were published after 2018, reflecting the actuality of and interest in port noise issues. Table 1 represents the analysis of the most relevant studies.

**Table 1.** Literature review.

| Element | Approach | Sample Port(s) | Output | Source |
|---|---|---|---|---|
| Port Noise | Ship noise | Analysis of regulatory framework | General | Measurement protocol | [32] |
| | | Mitigating noise impact | General | Measurement protocol | [33] |
| | | Predicting noise | Venice | Protocols and numerical model | [34] |
| | | Noise measurement | Rotterdam and Amsterdam | Possibilities of shore power installation and effect | [35] |
| | Port area | Noise mapping | Koper | COVID-19 impact overview | [27] |
| | | | Athens agglomeration | Noise maps and action plans | [6] |
| | | | Los Angeles | Predict noise levels | [36] |
| | | Noise monitoring | General | Source detection and noise map | [37] |
| | | Influence of meteorological conditions | Thessaloniki | Simulations of environmental noise levels | [38] |
| | | Low-cost tools usage | Naples | Improve the quality of noise monitoring | [39] |
| | | Residential exposure | Dublin | Implementation of the appropriate noise management and mitigation measures | [28] |
| | | Airborne noise | Naples | Prediction tool (simulation) | [40] |
| | | Ships "en route" | Livorno | Acoustic propagation models | [22] |
| | | Acoustic camera utilization | Genoa | Proposal of novel framework | [13] |
| | | Evaluate and compare noise standards | TanjungPriok | Measures for improvement | [41] |
| | | Noise monitoring and evaluation | Patras and Tripoli | Homogeneous best practices and procedure guidelines | [42] |
| | Port system | Institutional perspective and comparison with stakeholder reports | Spain | Environmental sustainability planning measures | [43] |
| | | Survey on monitoring systems, previous measurement campaigns, noise maps, and citizens' complaints | Ports of the North Tyrrhenian Sea | Stakeholder feedback | [29] |
| | | Noise source classification | Mediterranean | Effective noise management | [8] |
| | | Questionnaire analysis | Italian ports (selection) | Survey improvement | [44] |
| | | Overview of most relevant projects | North Tyrrhenian sea | Upgrade technical knowledge | [45] |

Considering the nature of the research, almost the entire analyzed portfolio of relevant studies confirmed the significant effects of port noise and identified the deficiencies of the regulatory framework, which act as constraints to effective noise management from numerous sources in the port area. As for the diversity of publications that had distinctive assumptions, they naturally generated multifarious conclusions. The vast majority of the analyzed literature related to research in the Mediterranean area, and the remaining publications dealt with noise pollution in western Europe, the western coast of the US, and Southeast Asia. Noise-mapping has been recognized as crucial for the effective determination of noise sources and their impact on the port area. In addition, the authors emphasized the need for an official measurement protocol in order to standardize the measurement and validation of port noise from an operational standpoint.

### 3. Materials and Methods

The purpose of this study is threefold: (a) to classify the noise sources in the port originating from two sources, traffic and industrial, (b) to provide an overview of the strategic noise maps, and present theoretical simulation(s) of noise propagation from ships at berth according to the determined propositions, and (c) to calculate the external costs of noise pollution due to the social and environmental damage generated in the subject area. Figure 1 shows the flowchart of the study.

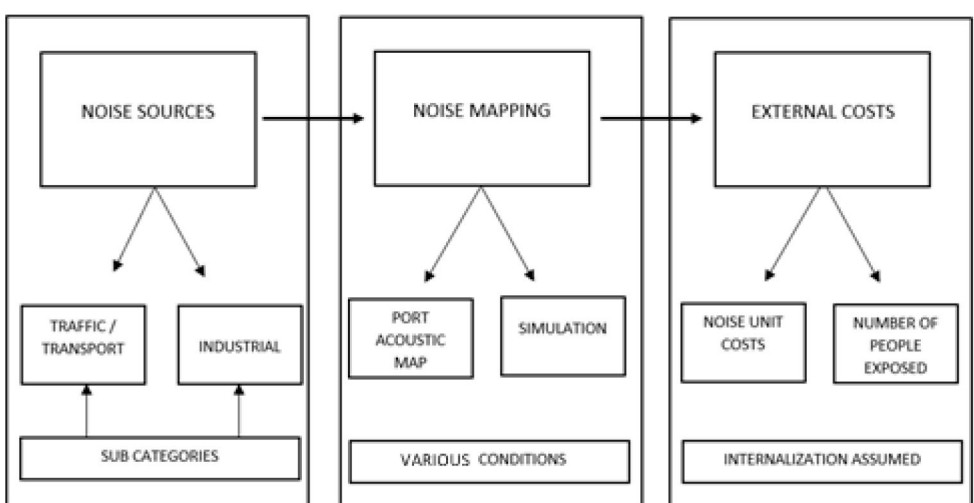

**Figure 1.** Flowchart of the study.

The research deals with acoustic pollution in the port of Split, focusing on the cargo terminals. The port of Split is the largest passenger port on the eastern Adriatic, and is located in the city of Split (Croatia), which is part of the Central Dalmatia region. The specific location, and especially the configuration, of the port contribute to the complexity of noise propagation. The facilities in the southern part of the port are on both sides of the semi-enclosed and narrow port basin. In addition, the port being located near an inhabited area surrounded by residential buildings and private houses increases the risk of inhabitants being exposed to increased noise levels at peak periods of everyday operation. This generates a specific acoustic environment, with a need to apply classifications to individual noise sources in order to create acoustic maps and calculate noise damage costs. Figure 2 represents the cargo port basin in the Split port with a cross-section of the purpose of its inherent parts. The highlighted zone represents the research area of this paper.

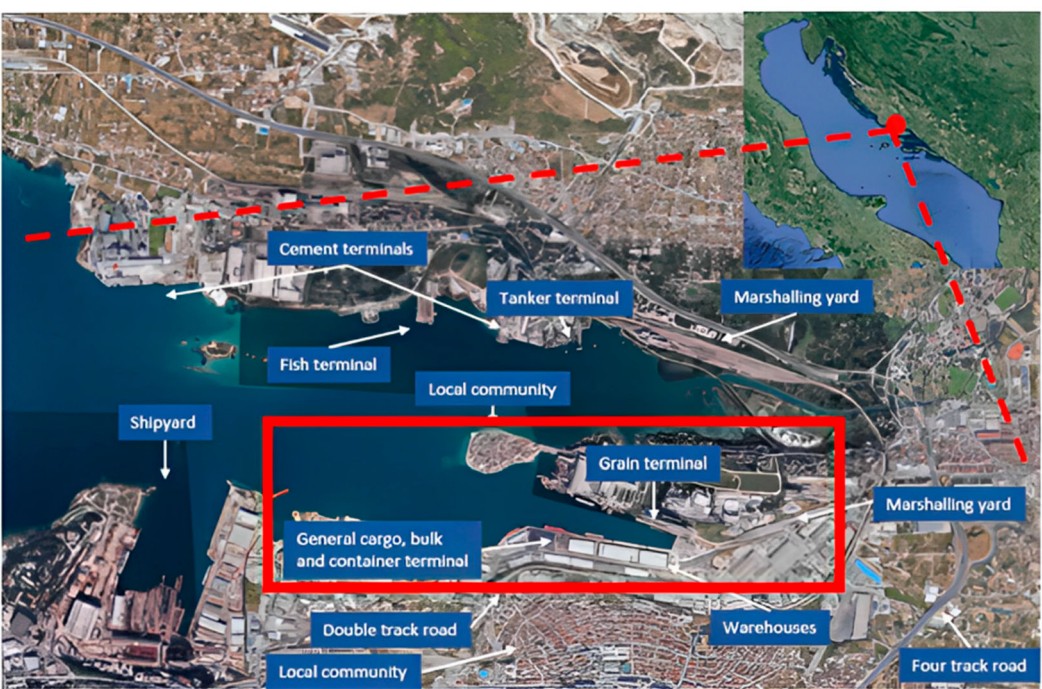

**Figure 2.** Overview of the cargo basin and infrastructure with indication of its functions ([46], modified).

### 3.1. Segmentation of the Port Noise Sources

The noise sources can be primarily categorized as traffic or industrial [26]. These categories represent the broader aspects of activities in a port, containing different individual components that should be analyzed holistically to determine their contribution to overall noise damage. The characterization of single noise sources in the study location can be elaborated following the methodology reported in Fredianelli et al. (2021). This approach divides the noise sources into five macro-categories as follows: road, railway, ship, port, and industrial. During noise-source determination in ports, it is important to provide a distinction between the traffic noise originating from activities inside the port area and those generated outside the boundaries of the observed port basin. Thus, in dealing with the noise sources, the research area's physical extent needs to be defined thoroughly in order to allocate the responsibility for generated noise in the complex geographical environment and to apply the proper mitigation measures. The strict separation of traffic-related sources is occasionally sorely missed, as sometimes port and urban traffic activities are merged. This demonstrates the complexity of port noise source segmentation and their impact on the environment and the local community. According to suggestions from Fredianelli et al. (2021) [8], traffic-related noise in ports includes internal, external, and external (urban) traffic dissociated from the port area. The industrial noise in the port area is related to the remaining port activities that are usually a constituent part of its functions and geographical position, comprising manufacturing activities, shipbuilding, free trade zones, workshops, and other industrial-related areas. During the first quarter of 2022, the monitoring campaign was performed to make a database of all the individual noise sources in the cargo terminal, including port surroundings. Database creation is an essential step in identifying critical points.

### 3.2. Noise Mapping and Simulations

A strategic noise-map is a tool that provides a geographical view of a particular area used to identify and prioritize actions for the reduction of noise-related sources which exceed predetermined threshold values. These maps usually comprise three daily intervals: day, night, and day–night–evening periods, according to the official ordinance [47]. The first step in noise-map creation is acoustic modeling of the examined area, based on the

composed input data used, to make a 3D digital model. The input data comprise the geographical data on the examined terrain; traffic data of road, rail, and industrial activities; data on the conditions of space utilization; meteorological data; population data; and others. The second phase of the process is the definition of individual layers of noise maps and model calculation, which culminates in strategic noise-maps as output data in the final phase.

Noise maps form the basis of providing action plans by implementing, and consequently combining, the data gathered from simulations in micro-space (specific port terminals) as an added value. Thus, the scope of the noise effects in the examined area increases. The simulation of the noise propagation of a ship at berth in the predetermined conditions is an effective tool to identify the areas of concern with noise intensity above the criterion level. According to the inverse square law, the sound intensity decreases by 6 dB for each doubling of sound distance under ideal conditions [48]. This data, with a maximum noise level of 55 dB in open space for the zone of mixed-, predominantly residential, use in the vicinity of the cargo terminals—which was prescribed by the state— were the main assumptions made for the initialization of simulations. For this study, two hypothetical scenarios regarding the cargo terminals of the port of Split were determined, considering the following measurement parameters: distance from the microphone to the noise source (ship and cargo terminal), sound level in dB, and overall noise propagation distance (from the origin to the open space). It should be emphasized that the sound levels used in the simulations were based on values determined in accordance with best practices in the academic literature and the port community. In addition, the inverse square law was considered when providing Scenario 2, corrected for external factors. Table 2 provides the parameters applied in the simulations.

**Table 2.** Parameters considered for the simulations of noise propagation on the cargo terminals in the port of Split *.

| | Noise Sources | Distance from the Microphone to the Noise Source | Sound Level | Distance of the Noise Propagation from the Source to the Open Space | Criterion Level |
|---|---|---|---|---|---|
| Scenario 1 | Funnel | 1 m | 110 dB | 562 m | |
| Scenario 2 | Ship cumulative (funnel, auxiliary engine, ventilator, etc.) | 40 m | 70 dB | 225 m | 55 dB |

* inverse square law considered.

The authors assumed ideal conditions as the theoretical propositions of the port terrain, such as a flat surface without obstacles, e.g., objects, buildings, warehouses, cargo-handling equipment, traffic infrastructure, and other resources used for traffic and industrial activities. In a later stage, simulation results will be compared with real-world measurements and used to formulate the main conclusions. When performing on-site measurement—due to the configuration of the cargo terminal and distances between residential buildings and port infrastructure—it is necessary to consider all the physical obstacles contributing to the reflection, refraction, diffraction, and absorption of noise, as well as other facts which possibly influence the final results. On-site measurement is the principal research procedure. The noise measurement protocol for moored ships was the output and guideline of the NEPTUNES project aiming to standardize the measurement process of airborne noise emission in ports, so future measuring should comply with the best-practice instructions [33]. Figure 3 represents the complex geography of the cargo terminal in the port of Split, surrounded by residential buildings in the southern part and the inhabited peninsula in the northwestern one.

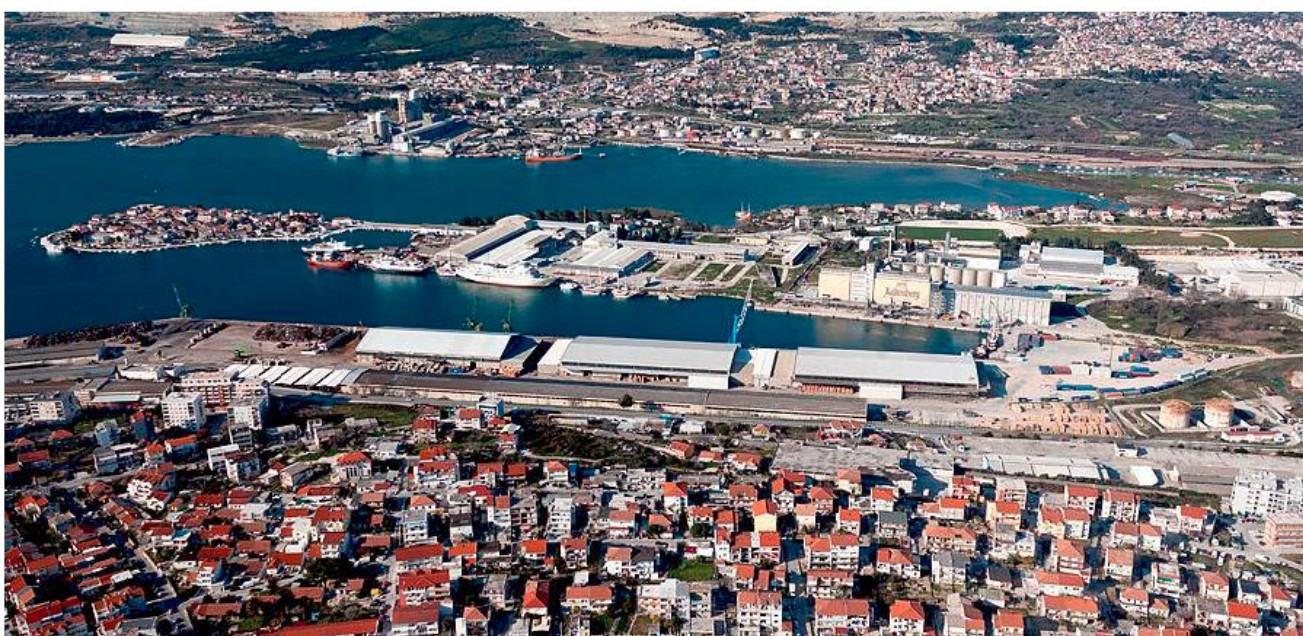

**Figure 3.** Aerial view of the cargo terminal in the port of Split [49].

*3.3. Methodology for Monetization of Noise Environmental Damage*

The monetization of external noise costs which are compensated by society, including local inhabitants situated on the port area rim, is a significant indicator of the magnitude of noise effects exceeding the prescribed values. There is a strong correlation between an increase in port traffic and noise emissions reaching adjacent buildings, the location of which is, in some instances, an omission of urban planning. The methodology for calculating external noise costs was based on the most recent handbook by van Essen et al. (2019) [21]. The input data comprise two values, the number of people exposed to noise and noise costs per exposed person. The latter can be further divided into annoyance costs and health costs. The calculation is expressed as follows:

$$EC_t = E_j \times NC_j, \tag{1}$$

where $EC_t$ is the total external noise costs from port t, $E_j$ is the number of people exposed to noise in the inhabited area j, and $NC_j$ is noise costs per person exposed in the inhabited area j.

More than 30% of European people are exposed to road noise and 10% to rail noise of intensity above 55 dB [50]. Most people are irritated by noise levels above 40 dB, while noise levels above 55 dB seriously annoy. At noise levels of 50 dB, most people feel moderate nuisance with sleep disorders [51]. Regarding the total number of the parameters involving the exposed population, the threshold above which noise is considered a nuisance was set to 50 dB, and classes of people exposed to noise were apportioned in bins of 5 dB. According to the methodology, that was necessary to keep the calculation process and input definition comparable. The number of people exposed to cargo port noise has been determined by combining the value of the total number of inhabitants and the estimated number of people affected by excessive noise levels. Table 3 shows the number of exposed people taken into consideration for calculation of external noise cost in the districts of two separate places, as the area on the northern part of the terminal is within the borders of the town of Solin and the southern part belongs to the jurisdiction of the city of Split. The estimated percentage of the exposed population was determined in accordance with the results of the on-site survey conducted as an outcome of the following project activity.

**Table 3.** Total number of people exposed to noise from cargo terminals in the port of Split.

|  | Total Population | Percentage of Exposed Population | Total Exposed People |
|---|---|---|---|
| Split city district—Brda | 6188 * | 60% | 3713 |
| Solin town district—Vranjic | 1066 ** | 70% | 747 |

* Ref. [52]; ** Ref. [53].

The remaining input data, unit costs of environmental traffic noise, consider the loss-of-welfare that arises from an extra decibel of increased noise [54]. The environment price values, composed of the annoyance and health costs, were based on the data indicated in the handbook of van Essen et al. (2019) [21] and are expressed as:

$$NC_j = A_p + H_p,\tag{2}$$

where $NC_j$ is noise costs per person exposed in the inhabited area j, $A_p$ is the annoyance cost per exposed person p, and $H_p$ is the health cost per exposed person p.

The noise costs in maritime transport were neglected in the study, implying that the same approach was also applied to all four operational phases of a ship, including the berthing of a ship in port. Lacking relevant data on the environmental price of port noise, the authors assumed that road and rail traffic and their respective values are closely related to the noise generated in ports. This was a consequence of taking the port into consideration as a transport node where vehicles, railcars, and ships operate and jointly contribute to the detrimental effects of the overall port noise. The aviation traffic noise values were not applicable due to the specific characteristics and environment. Table 4 provides the environmental price of road and rail traffic noise, cumulative for EU28.

**Table 4.** Environmental price of road and rail traffic noise combined to determine port noise values for EU28 (2016 EUR/dB/person/year) [21].

| $L_{den}$ (dB(A)) | Road Transport | | Rail Transport | | | |
|---|---|---|---|---|---|---|
| | Annoyance | Health | Annoyance | Health | Annoyance | Health |
| 50–54 | 14 | 3 | 17 | 14 | 9 | 17 |
| 55–59 | 28 | 3 | 31 | 28 | 4 | 32 |
| 60–64 | 28 | 6 | 34 | 28 | 6 | 34 |
| 65–69 | 54 | 9 | 63 | 54 | 3 | 63 |
| 70–74 | 54 | 13 | 67 | 54 | 13 | 67 |
| >75 | 54 | 18 | 72 | 54 | 18 | 72 |

As the data was segmented according to the individual sound level for the interval $L_{den}$ (day, evening, night) (A-weighted and accompanied values), the authors calculated the mean of all data to maintain homogeneity. The premise was to determine the minimum level of port noise prices as a realistic overview of noise effects in the examined area, even though the real-world values could be much higher, in order to contribute to the sustainable development paradigm.

For calculation of external noise costs, the authors have presumed the complete internalization of external costs in the transport sector. This implies the generated pollution is monetizable and that there is an obligation on the part of the service provider or polluter to pay for the caused damage.

### 3.4. Noise Maps in the Port of Split

The strategic noise maps were retrieved from Štimac et al. [55] and related only to the southern terminals of the port of Split, as the northern cargo terminals belong to the other municipality. Concerning the monitoring campaign performed, the noise levels recorded

on the southern terminals could also be usable for northern facilities as for the similarity of the terrain, noise emission levels, and surrounding area. The noise maps were determined for road, rail, and industrial noise sources for the evaluation periods: day ($L_{day}$), evening ($L_{evening}$), night ($L_{night}$), and day–evening–night ($L_{den}$). Each noise graph indicates equal classes of noise levels. Between adjacent classes, 5 dB(A)-wide bands were marked starting from 45 dB(A) to levels above 80 dB(A).

According to set data, the following figures show strategic maps of cargo terminals in the Port of Split. Figures 4 and 5 indicate a strategic road and rail transport map in selected four evaluation periods.

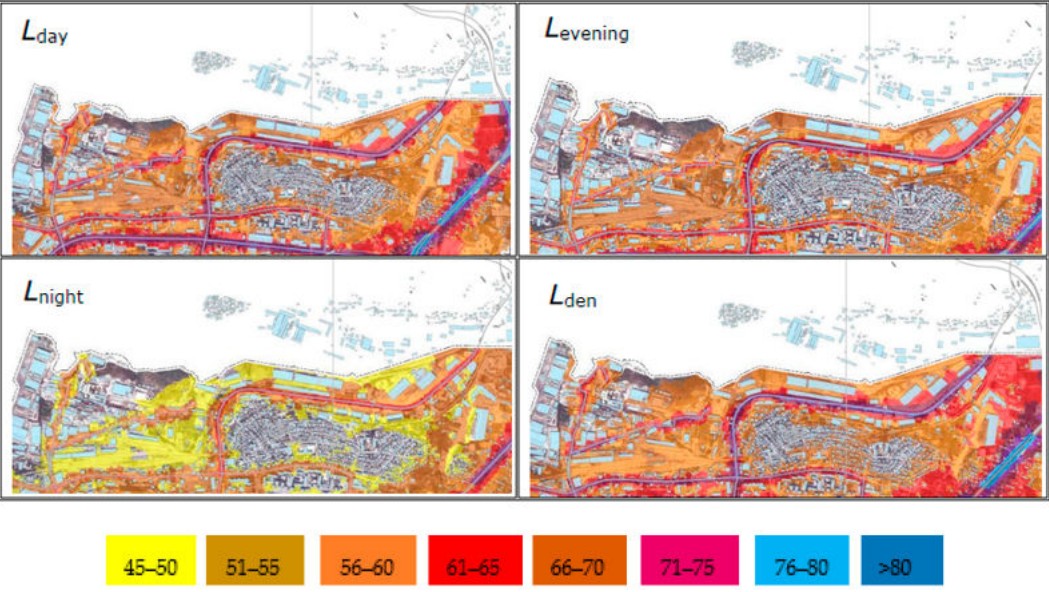

**Figure 4.** Strategic maps of road noise sources in the evaluation periods ($L_{day}$, $L_{evening}$, $L_{night}$, $L_{den}$) * [55]. * noise indicator class (dB).

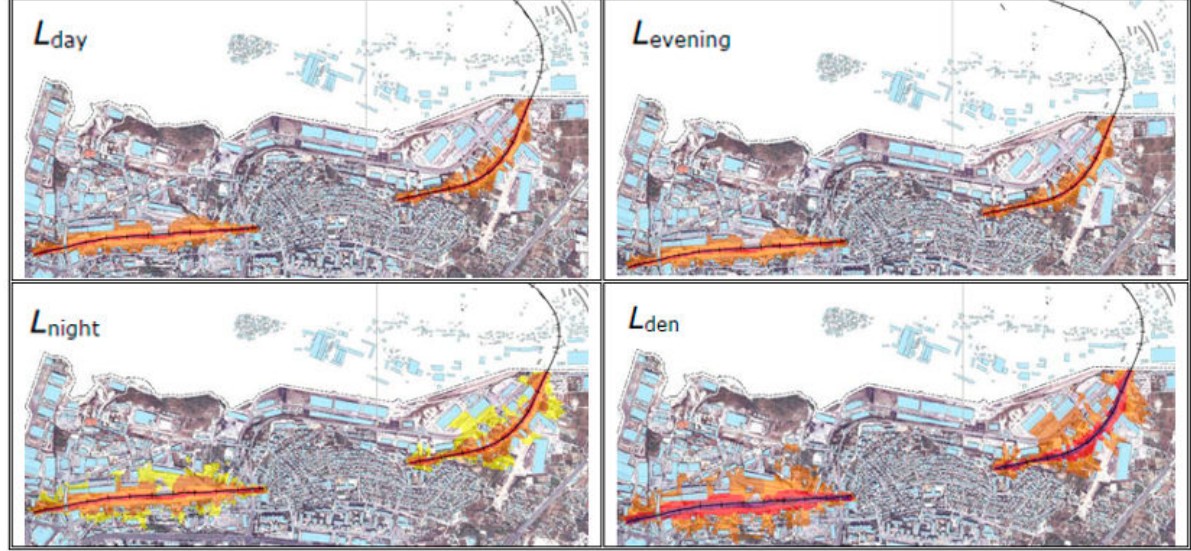

**Figure 5.** Strategic maps of rail noise sources in the evaluation periods ($L_{day}$, $L_{evening}$, $L_{night}$, $L_{den}$) * [55]. * noise indicator class (dB) as in Figure 4.

Finally, Figure 6 provides a strategic map for industrial facilities, including the cargo terminal, in the four evaluation periods selected.

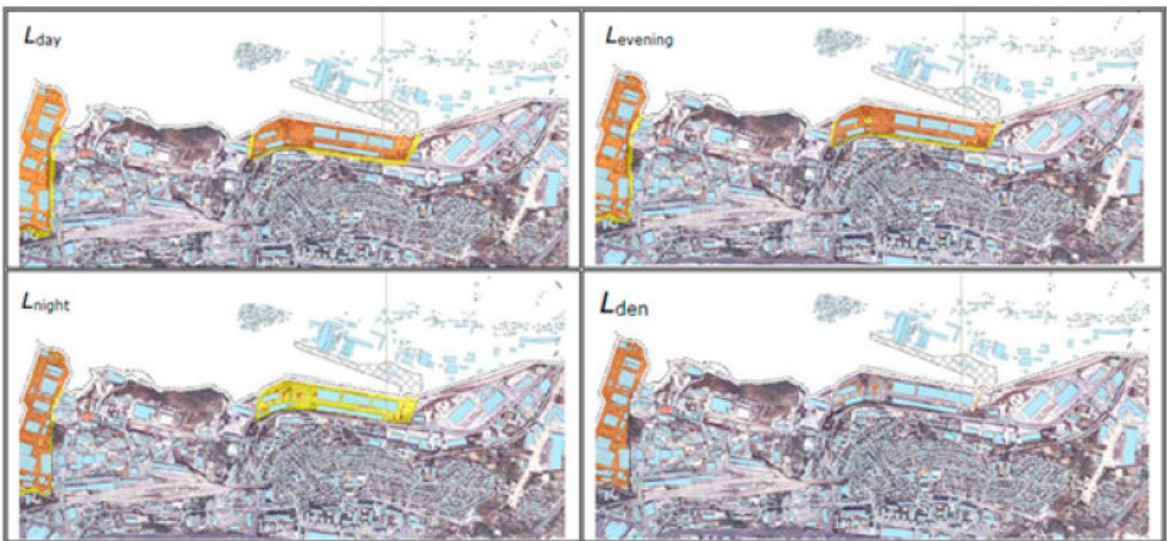

**Figure 6.** Strategic maps of industrial noise sources in the evaluation periods ($L_{day}$, $L_{evening}$, $L_{night}$, $L_{den}$) * [55]. * noise indicator class (dB) as in Figure 4.

## 4. Research Results

The results of the research objectives, broken down into three separate fragments, are presented in the following paragraphs.

### 4.1. Port Noise Sources

The first fold of the research was noise source classification at cargo port terminals in the port of Split. The authors created a database of the main road, railway, ship, port, and industrial sources by monitoring traffic during the daily port operations. In general, the cargo terminals being surrounded by residual facilities and the heavy freight flows regularly mixing with urban traffic contribute to higher background noise sound levels. An illustrative overview of the main categories of environmental noise sources is presented in Figure 7.

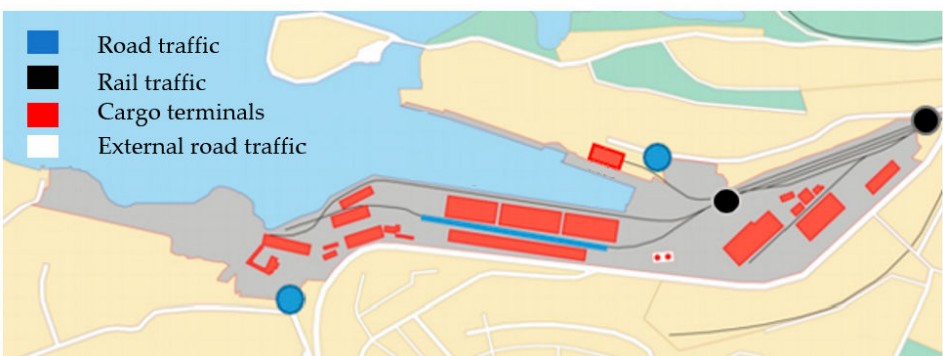

**Figure 7.** Main noise categories in the research area of Split port ([56], modified).

The road and rail sources were divided into three components, as discussed in the previous sections. Internal road traffic is related to the entrance gate flows (in blue circles) and internal port-road traffic flows (blue line). Analogously, the rail noise sources in the port of Split consist of the entrance points of railway infrastructure (black circles) and remaining internal rail tracks, which penetrate directly to the berths (black lines). The outcome of the internal port infrastructure of these two traffic modes being set jointly at some points, even parallel, is higher noise levels. The entrance gates are interface boundary points between internal road-traffic port noise and port-related external traffic, which—in this case—are access roads to the cargo terminals. In periods of higher traffic intensity,

these access roads are partially congested. Finally, external road traffic (white line in the southern part behind the warehouses) is always intensive with high density during daily peak periods. Generally, this is a single-track road in both directions and is highly congested—generating significant and constant noise with daily variations—especially as there is no direct connection to the highway. The night–evening periods are much less intense, and the road noise sources produce lower but continuous noise levels. As regards port-related external rail traffic, the marshaling yard is located on the eastern part of the port area, where several rail cargo shipments are encountered. Additionally, external rail traffic consists of a second track running towards the passenger terminals in the city center. It slightly annoys the local population, mainly in the evening. These two rail tracks are separated by a railway fork located ahead of the marshaling yard, so the noise effects of these two tracks are distinctive. External rail traffic in the port is negligible, as the port is the last point on the line ranging towards outbound areas. Table 5 shows road and rail noise sources registered in the port of Split.

**Table 5.** List of the main road and rail traffic noise sources in the port of Split.

| | Internal Traffic | Port-Related External Traffic | External Traffic (Not Generated by Port) |
|---|---|---|---|
| Road | LGV | Motorcycles | Motorcycles |
| | HGV > 32 t | Passenger cars | Passenger cars and vans |
| | Industrial vehicles | Vans | Tourist vehicles |
| | Mobile yard cranes | Light commercial vehicle (LCV) | Light commercial vehicle (LCV) |
| | Reach stackers (container) | LGVs and HGVs | Bus/coach |
| | Forklifts | | Special industrial vehicles |
| Rail | Locomotives | Marshaling yard | LGVs and HGVs |
| | Railcars (mainly bulk) | | Second track leading to the passenger terminals |

Ship noise in ports implicates emissions from several sources which reproduce intensive low-frequency noise, and thus, cause annoyance to residents. It is mainly related to the exhaust funnel of auxiliary engines, ventilation openings, and pumps. This noise is generated by the continuous operation of ship machinery systems while staying in port and generally has tonal and intermitting noise characteristics. Nuisance effects are expressed particularly during evening and night, as is the case for a low density of urban traffic and relatively higher effects of noise generated by moored ships. However, the variations in noise levels occur according to ship types, ship specifics (LOA, beam), duration of stay, and cargo type (bulk, container, and special cargo). Table 6 provides the data on the most frequent vessels calling to the port of Split and related specifics, segmented on several criteria, which contribute mutually to higher noise levels.

**Table 6.** Ship specifics according to the several criteria in the port of Split (maximum values).

| Ship Type | DWT | LOA | Beam | Duration of Stay |
|---|---|---|---|---|
| General cargo ship | 15,000 | 140 m | 20 m | 3–4 days |
| Bulk carrier | 62,000 | 200 m | 33 m | 5 days |
| Container vessel (feeder) | 13,000 (950 TEU) | 140 m | 20 m | 1 day |
| Oil/chemical tanker | 6000 | 110 m | 18 m | 5 days |

The additional increase in noise levels can result from tug-handling activity, obligatory in specific micro-climate conditions, and pilotage. Furthermore, port noise sources primarily originate from quayside cargo-handling equipment, pure handling of an individual type

of cargo, and warehousing activity. The port of Split is equipped with several portal cranes for general cargo-handling, directly on the interface of quayside and ship. In addition, the port is multi-purpose, manipulating various cargo types, from scrap metal, bulk cargo, and special cargo (windmills) to containers. Cargo-handling activity usually causes impulsive noise, as opposed to stable, continuous noise generated by a berthed ship. The level of noise emission from the port depends on the handled cargo specifics and period of day (day–evening–night). The noise of port traffic and port noise regularly interfere, so strict differentiation is sometimes doubtful. Finally, the industrial noise sources in the port area include manufacturing, assembly, warehousing, shipbuilding of small vessels and crafts, and others. Taking place in open space, it can generate continuous values in excess of limits—and thus, cause health-related effects for the local population—as opposed to operations in closed objects or warehouses. Table 7 provides the list of port and industrial noise sources in the port of Split.

**Table 7.** Port and industrial noise sources in the port of Split.

| Port Noise Sources | | Industrial Noise Sources |
|---|---|---|
| **Port Equipment** | **Cargo Type** | |
| Ship loaders (bulk cargo) | Scrap metal | Silos for grain storage |
| Belt conveyors | Containers | Closed warehouses for various activities (manufacturing, cargo storage) |
| Elevators | Bulk cargo (slag, cooper ore, calcite, grain) | Semi-open warehouses |
| Hooper | Special cargo (windmills) | Shipbuilding of small vessels |
| Quay cranes | Liquid cargo (bitumen) | Workshops |
| Other cargo-handling equipment | | |

### 4.2. Simulations

Complementary to the analysis of strategic noise maps for selected sources and evaluation periods, the authors conducted a simulation of two scenarios to determine the noise propagation of a ship at berth in the predetermined conditions, which were assumed ideally. The authors simulated the ship noise propagation on every disposable berth on the cargo terminals. These simulations also included the cement and tanker terminals, which were not initially in the scope of the research objective. This was performed to determine the complementary impact of terminal noise on the Vranjic peninsula and other affected areas. Scenario 1 assumes the onboard noise measurement directly from the source, considering the inverse square law and noise level of 110 dB. The first scenario simulation is indicated in Figure 8.

Analogously, the propositions of Scenario 2 are based on noise measurement on the terminal at a certain distance (40 m) with an initial noise level of 70 dB. The simulation of the second scenario is shown in Figure 9.

The effects of the presented simulations show differences in the affected areas, i.e., where the exceeding of the noise threshold levels occurs within the zone of mixed-, predominantly residential, use. They are shown in the form of concentric circles. By considering the number of exposed inhabitants to the excessive noise levels from moored ships in the port of Split, the simulations point to clear conclusions when considered with the premise of ideally set propositions.

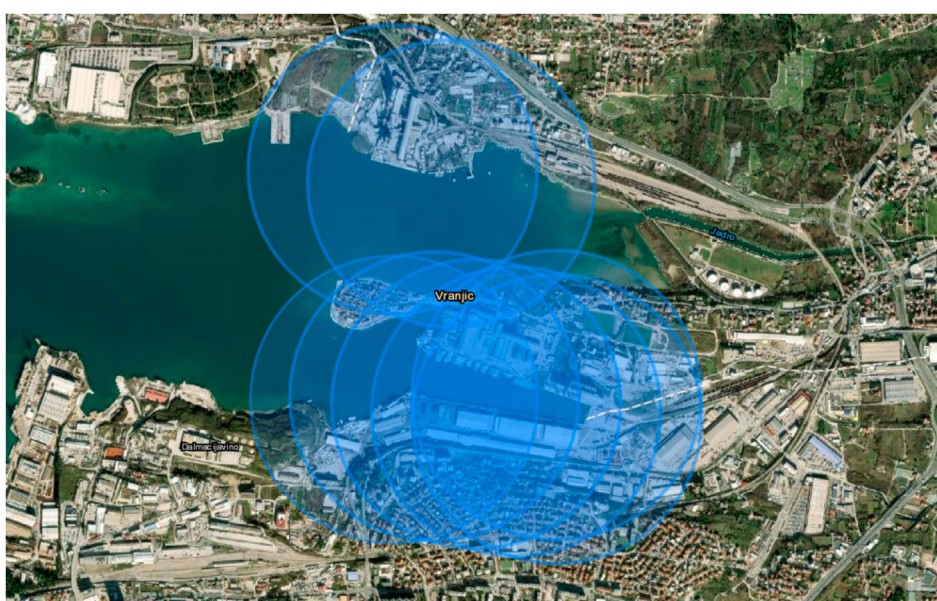

**Figure 8.** Simulation of Scenario 1 ([46], modified).

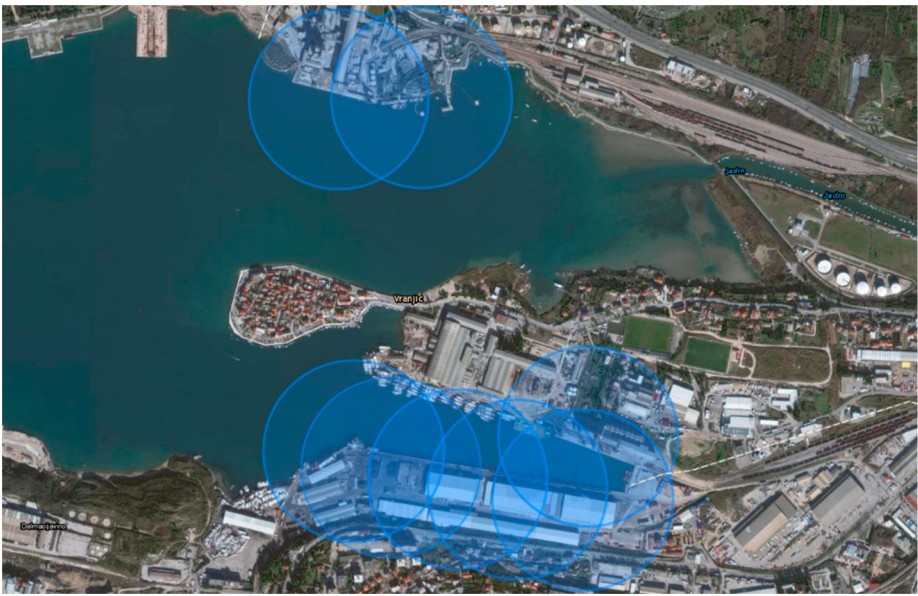

**Figure 9.** Simulation of Scenario 2 ([46], modified).

### 4.3. Calculation of External Noise Costs in the Port of Split

By analyzing the noise maps of selected noise sources (road, rail, and industrial), it can be assumed that the dominant influence on noise by a day–evening–night indicator ($L_{den}$) varies, ranging from 55 dB to 60 dB. According to the official regulation [57], the maximum permissible noise levels for the $L_{den}$ indicator in the examined area are 57 dB. Based on the arithmetic mean of road and rail traffic noise values for annoyance and health costs indicated in Table 4, the value of 28 EUR/person/year was taken as an annoyance referent cost and 3.5 EUR/person/year for health costs. These data were multiplied by the number of exposed people in the examined area to determine the total external noise costs from the port area. Table 8 shows the recapitulation of yearly external noise costs in the port of Split.

**Table 8.** Total external noise costs in the port of Split (yearly levels).

| | Annoyance Costs | Health Costs | Number of People Exposed | Total External Noise Costs |
|---|---|---|---|---|
| Split city district—Brda | 28 EUR/person/year | 3.5 EUR/person/year | 3713 | EUR 116,960 |
| Solin city district—Vranjic | 28 EUR/person/year | 3.5 EUR/person/year | 747 | EUR 73,206 |

The total external noise costs in the cargo port of Split were calculated to be 190,166 EUR/year, considering the number of exposed inhabitants and assumed noise levels.

## 5. Discussion

The airborne noise emissions from ports—usually situated in urban areas—generate various health-related influences such as anxiety, stress, cardiovascular diseases, fatigue, and similar nuisance effects. In order to control airborne ship noise emissions, the Lloyd's Register published a new airborne noise emission notation that limits ship noise emission levels in ports [58]. Identification of individual noise sources and classification is a central part of a noise abatement program [27]. In this research, continuously monitoring of the noise sources during everyday port activities was performed. These sources were segmented according to their physical presence and operation in the port area and categorized into four macro-categories: road, railway, port, and industrial sources. The purpose of noise source segmentation was to utilize the obtained data to create a usable database for noise mapping in future phases. A more comprehensive segmentation of each category of noise source can be found in [8], which confirms the importance of identifying the critical points in the noise management of the port area. By identifying the individual polluters, a priority-minded source list of actors that most exceed the permissible noise levels can be developed. This enables decision-makers and port-managing entities to adopt proper and effective noise-management protocols to tackle incremental pollution. The road and rail traffic noise sources were further divided into subcategories, as the internal, port-related external, and external sources have different noise features. Noise maps analysis indicates the most influential noise traffic sources originating from road and rail operations externally and occasionally exceeding the limit values. According to the current regulations, ship noise sources in the port area, closely related to the different types and specifications of ships at berth, were not considered in simulations. They will be included in the measurement protocols of the following research phase to show their share of the total noise in the port. The primarily sources of ship noise on the berth are auxiliary engines, ventilator openings, pumps, generators, and others, which can increase the noise intensity in the area. The noise of the berthed ship is continuous with tonal characteristics. The noise emission from maneuvering mode of ships was excluded from the analysis, as they are hardly traceable. Port noise is generated primarily from the quay-side cranes and other stationary cargo-handling equipment. The physical features of the cargo handled in the port certainly contribute to the temporary increase in the total noise levels, having impulsive noise components. These noise effects are also combined with traffic activities and other industrial facilities in the port area, providing difficulties in identifying the origin of the exceeded noise levels and separating the single contributions. Due to the increasing awareness and importance of airborne port noise, and based on the same premises as in this work, the paper of Schiavoni et al. (2022) [59] confirms the results of this study and provides additional value in the form of database of sound power levels and spectral data of noise sources in the port area. It represents a significant improvement in the general knowledge of the contribution of each port noise source to the overall noise levels and enhances the standardization of measurements.

In defining the significance of noise sources' impact on the exposed local population, the collected data is generally used to create strategic noise maps. These maps provide noise emissions of the examined area identifying the most influential sources of noise

emission from traffic and industrial activities. By analyzing the available strategic noise maps of road, rail, and industrial noise sources with a focus on the surrounding port area, the noise originating from road and rail activities was the most influential during the day ($L_{day}$) and evening ($L_{evening}$) periods. The traffic noise often exceeds the permissible levels and thus reproduces nuisance effects towards the inhabited area. It refers to the external road located in the southern part of the cargo terminal with heavy traffic and congestion generating noise levels above 70 dB. However, the internal port traffic and port-related external traffic remain within the boundaries of acceptable levels. The same elaboration can be expanded to the traffic noise source analysis in the night period ($L_{night}$) but with lower exposure, variations, and, principally, noise levels. The noise generated from industrial sources (cargo terminal) during the day ($L_{day}$), evening ($L_{evening}$), and night ($L_{night}$) periods had acceptable noise emission levels in all analyzed components. Finally, according to the data provided in the noise maps, the noise levels of the day-evening-night period ($L_{den}$) were excessive in road and rail traffic, as opposed to the noise emission from cargo terminals, which were negligible. The outcomes of noise maps analysis are essential for the noise management protocols adoption to handle excessive noise immission levels in urban facilities. In addition, the analysis point to the road and rail traffic sources rather than port noise as the most significant contributors to the environmental and social adverse impacts. The noise propagation simulation from the cargo terminals was performed to determine the noise emission when modeled in specific conditions. The authors assumed the ideal port area conditions in particular surroundings, which included the absence of all the objects and obstacles that could interfere with sound propagation. It was performed to compare the simulation results with real-time data in later phases. The simulations with two scenarios suggest the noise emission from the berthed ship and the accompanied area of exposure, which determines the zone of excessive noise. Two simulation scenarios differed considering the distance of sound measurement, including the inverse square law as a primary assumption. Finally, the ultimate objective was to calculate the damage noise costs of the examined area, with the premise of complete internalization of external costs, as a mechanism to monetary express the caused damage to the environment and the local community. The procedure combines the annoyance and health costs with estimating the number of exposed people to excessive noise levels. The authors estimate a total of 190,166 €/year of damage costs from the noise sources surrounding the considered populated areas. These data are usable in future noise management plans aiming to lower the noise impacts from noise sources in the vicinity of inhabited regions.

This multi-faceted study presents a holistic approach to conducting the impact analysis of the noise emission from the cargo terminals in the Split port. The research integrated three inherent components, which logically show the current state of the noise effects and its impact on populated areas. This research data should implore the relevant institutions to strengthen noise-management plans, and effectively and continuously monitor noise emission in critical areas. It has particular importance in the peripheral areas where industrial and urban activities interact. Also, it is necessary to include the industrial activities' projected impacts, terrain purpose, and the specificity of local community areas in future urban planning processes. The outcomes of this research also emphasize the need to adopt a standardized measurement procedure with precise evaluation of noise sources. This was identified as one of the most important requirements from the literature analyzed in Table 1. It would increase the reliability of the data generated. In addition, the prerequisite for strategic noise map comparison is the definition of strict guidelines, focusing on the specifics of the port area. Due to the health effects, the environmental noise generated from port activities should be included in the official regulations and policies as a separate category. The latter would increase the importance, awareness, monitoring, and management of the port noise's detrimental effects.

## 6. Conclusions

Throughout this research, the authors identified principal noise sources in the port of Split, assessed the intensity, simulated the propagation, and calculated the annual noise costs due to the negative impact on residents of nearby populated areas. It should be emphasized that in developing this research, the authors used some assumptions which can be considered study limitations. The authors identified the noise sources in the port and retrieved the acoustic maps from the available general documentation. Due to the lack of correct data, the unit noise costs were determined and used as mean values. All these procedures, carried out to identify noise sources and simulate noise propagation in the port area, are used to prepare the data generated for the measurement campaign, determining what and where to measure. In the following phases of this integral project, the authors will evaluate the value of the data presented by comparing it to real-time data.

**Author Contributions:** Conceptualization, L.V.; methodology, L.V. and I.P.; validation, I.P.; formal analysis, L.V. and R.G.; resources, R.G.; data curation, L.V.; writing—original draft preparation, L.V.; writing—review and editing, I.P. and R.G.; visualization, R.G. All authors have read and agreed to the published version of the manuscript.

**Funding:** Research activities presented in this paper were conducted under the scientific research project "Measurement and validation of external airborne noise from ships in the port of Split (01/2021)" supported by the University of Split Faculty of Maritime Studies, Croatia.

**Institutional Review Board Statement:** Not applicable.

**Informed Consent Statement:** Not applicable.

**Data Availability Statement:** The data presented in this study are available on request from the corresponding author.

**Conflicts of Interest:** The authors declare no conflict of interest.

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
