# Peer review of "Multi-Faceted Analysis of Airborne Noise Impact in the Port of Split (I)"

_jmse, doi:10.3390/jmse10101564_

Round 1
Reviewer 1 Report
Reviewer Comments to Author(s):
It is very interesting to study the impact of external airborne noise in ports and to investigate the sources of noise present in these environments. The paper is generally well written and structured. However, in our opinion, the paper has some shortcomings in some sections. For this reason, this study has the potential to be accepted after a major review.
These points can be summarized as follows:
-
Lines 134 to 141 of the text, which are regarded as a description of the article's structure, should be deleted, and there should be replaced with a simple description of the study's objectives or main research questions.
-
Change the word "Authors" to "Source" in Table 1.
-
Analyze the information presented in Table 1 in further detail, and then include (Table 1) in the text of the manuscript (lines 159-170).
-
Please describe in the text where the Split port is located; in which city, in which region, and in which country (lines 185-186). I suggest that the localization of the studied port be shown in Figure 2 as well, if at all possible.
-
In the same paragraph, Fredianelli et al. (2021) [8] is mentioned twice. Keep the first one (line 204) exactly as it is and just use the reference [8] per second (lines 215-218).
-
Improve the quality of Figure 4. I also think that this Figure needs a legend.
-
Remove Figure 5 from the manuscript and add it as a legend to Figures 6, 7, and 8.
-
Improve the quality of the maps shown in Figures 6, 7, and 8.
-
The discussion of your research is very general in terms of how you present the main findings; it is necessary to position your work in comparison with the scientific literature; furthermore, it is imperative to strengthen your interpretations by using recent sources.
-
I suggest you change the title to “An analysis of how external airborne noise affects port environments: a case study of Split port”
Reviewer 2 Report
Minor issue: the link to reference 17 was unavailable at the moment of the review: "The resource you are looking for has been removed, had its name changed, or is temporarily unavailable."
Major issues I see need to be addressed:
- it is unclear how the percentages of population exposed to noise levels was estimated (Table 3, pg. 9, ***) In a research paper those percentages should be based on something more than an estimation of the authors.
You state in section 3 that the paper will classify the noise sources in the port. What I can find in section 4 (Research results), table 5, is a list of noise sources, but it is very general. When discussing noise pollution, I find it important to have values of the intensity of noise sources at different moments of the day. How loud is the noise produced by the mobile yard cranes for instance? What are the peak values, how does that sound integrate in the whole context of the port sound pollution? That same question should be addressed for all types of noise sources. What type of instruments were used to measure the intensity of the noise sources and who was in charge of the measurements? The way I see it, the part with the noise sources as it is now, does not have a place in the Research results section.
In research results, point 4.2 it is stated that "the strategic noise maps were retrived from Stimac et all [54]"... I could not find the maps at that particular reference and I don't understand if you generated the maps according to noise levels that were monitored or you took the already generated maps from reference [54]. If the latter is true, and I believe it is, than I consider that those maps are not part of your research results.
Figure 9 and 10 are simulations of 2 scenarios. The graphical method used to display the results should be improved to be similar to the maps in Figure 6 to 8.
In scenario 1 you state you are using a value of 110 dB. Is it measured, under what circumstanced, what equipment you are using? If you assumed a noise level of 110 dB, respectively 70 dB in scenario 2, what are the consideration under you decided for these values? Do you have an on-site noise measurements in different points to validate the simulation results?
Table 8 shows the total external noise cost that is based on the percentages of people exposed to noise levels in the 2 districts. As this is also part of the research results, I find it of outmost importance to state the method you used to find those percentages, as stated before.
What I believe is needed:
- surveys in the 2 districts to show how affected the population is by the port noise and to validate the percentages of people you took under consideration
- noise levels recorder in the 2 districts to validate the same percentages
- noise levels recorder in the port area to generate the noise maps
-noise intensity for the sources stated in tables 5 to 7, how this noise levels are considered in the creation of the simulated scenarios
-noise maps of the simulated scenarios with levels of the noise emanating from the origin source to the adjacent areas (in colors to have a better understanding how the 110 dB dissipated and affect the regions discussed).
The maps should be corelated to the surveys to show the number of people affected by the noise.
Reviewer 3 Report
The authors present and discuss their work on impact analysis of external airborne noise in port.
The topic is within the scope of Ocean Engineering of JSME.
The work is systematic, complete and well organized. The results are convincing.
Some small details need to be adjusted.
Minor comments:
1) Line 286,287:Why is the threshold of noise set to 50dB? Please explain in detail.
2) Figure 4 and Figure 5 are not very clear, please modify them.
3) This paper describes impact analysis of external airborne noise in the port of Split. Do the authors have any constructive comments for other ports?
Round 2
Reviewer 1 Report
I thank the authors for taking most of my comments into consideration.
Author Response
The authors wish to thank the reviewer for the valuable comments and suggestions provided.
Reviewer 2 Report
I appreciate the clarifications submitted with your response and the supplementary references provided.
The only small issue I believe it can be improved is the form of the paper due to the explanations added, respectively:
-page 6: there is a lager gap as Fig. 2 is now on page (I believe it can be rearranged)
- page 11: the same issues as on page 6... a large gap after fig. 4.
page 15 - gap as Fig 9 is on page 16
page 16 - the title of Chapter 5 is at the end of the page, it should go on page 17
Author Response
The authors wish to thank the reviewer for the valuable comments and suggestions provided.
Considering the issues indicated below, the authors couldn't find the indicated gaps (the issues occurred for the different word versions?):
- there is no gap above or below Figure 2 (checked)
- between Figures 4 and 5 there is only one blank row, to divide the illustrations (checked)
- Figures 8 and 9 are on page 15 (checked)
- The title "Discussions and conclusions" is set at the beginning of the paragraph, on page 16 (checked).